# Inhibition of Autophagy Aggravates *Arachis hypogaea* L. Skin Extracts-Induced Apoptosis in Cancer Cells

**DOI:** 10.3390/ijms25021345

**Published:** 2024-01-22

**Authors:** Chia-Hung Tsai, Hui-Chi Huang, Kuan-Jung Lin, Jui-Ming Liu, Guan-Lin Chen, Yi-Hsien Yeh, Te-Ling Lu, Hsiang-Wen Lin, Meng-Tien Lu, Po-Chen Chu

**Affiliations:** 1Department of Surgery, Taichung Tzu Chi Hospital, Buddhist Tzu Chi Medical Foundation, Taichung 427213, Taiwan; richardtsai4885@yahoo.com.tw; 2School of Chinese Medicine & Graduate Institute of Chinese Medicine, China Medical University, Taichung 406040, Taiwan; hchuang@mail.cmu.edu.tw; 3Department of Chinese Pharmaceutical Sciences and Chinese Medicine Resources, China Medical University, Taichung 406040, Taiwan; wilfriedyeh85@gmail.com; 4Division of Urology, Department of Surgery, Taoyuan General Hospital, Ministry of Health and Welfare, Taoyuan 33004, Taiwan; mento1218@gmail.com; 5Department of Urology, College of Medicine and Shu-Tien Urological Research Center, National Yang Ming Chiao Tung University, Taipei 112304, Taiwan; 6Department of Obstetrics and Gynecology, Tri-Service General Hospital, National Defense Medical Center, Taipei 114202, Taiwan; 7Department of Cosmeceutics and Graduate Institute of Cosmeceutics, China Medical University, Taichung 406040, Taiwan; pisceschen860320@gmail.com (G.-L.C.); tina5073300@gmail.com (M.-T.L.); 8School of Pharmacy, College of Pharmacy, China Medical University, Taichung 406040, Taiwan; lutl@mail.cmu.edu.tw (T.-L.L.); hsiangwl@mail.cmu.edu.tw (H.-W.L.); 9Department of Pharmacy, China Medical University Hospital, Taichung 406040, Taiwan

**Keywords:** peanut skin, melanoma, colorectal cancer, anticancer, autophagy

## Abstract

The skin of *Arachis hypogaea* L. (peanut or groundnut) is a rich source of polyphenols, which have been shown to exhibit a wider spectrum of noteworthy biological activities, including anticancer effects. However, the anticancer activity of peanut skin extracts against melanoma and colorectal cancer (CRC) cells remains elusive. In this study, we systematically investigated the cytotoxic, antiproliferative, pro-apoptotic, and anti-migration effects of peanut skin ethanolic extract and its fractions on melanoma and CRC cells. Cell viability results showed that the ethyl acetate fraction (AHE) of peanut skin ethanolic crude extract and one of the methanolic fractions (AHE-2) from ethyl acetate extraction exhibited the highest cytotoxicity against melanoma and CRC cells but not in nonmalignant human skin fibroblasts. AHE and AHE-2 effectively modulated the cell cycle-related proteins, including the suppression of cyclin-dependent kinase 4 (CDK4), cyclin-dependent kinase 6 (CDK6), phosphorylation of Retinoblastoma (p-Rb), E2F1, Cyclin A, and activation of tumor suppressor p53, which was associated with cell cycle arrest and paralleled their antiproliferative efficacies. AHE and AHE-2 could also induce caspase-dependent apoptosis and inhibit migration activities in melanoma and CRC cells. Moreover, it is noteworthy that autophagy, manifested by microtubule-associated protein light chain 3B (LC3B) conversion and the aggregation of GFP-LC3, was detected after AHE and AHE-2 treatment and provided protective responses in cancer cells. Significantly, inhibition of autophagy enhanced AHE- and AHE-2-induced cytotoxicity and apoptosis. Together, these findings not only elucidate the anticancer potential of peanut skin extracts against melanoma and CRC cells but also provide a new insight into autophagy implicated in peanut skin extracts-induced cancer cell death.

## 1. Introduction

Natural products and phytochemicals have been critical sources of developing preventive or therapeutic drugs for several human diseases, especially infectious diseases and cancer [1]. From the therapeutic perspective, a number of plant-derived compounds, such as vinblastine, vincristine, paclitaxel, and camptothecin, have significantly contributed to the successful clinical treatments of many cancers [2]. Moreover, substantial evidence has demonstrated that dietary bioactive compounds found in fruits, vegetables, and whole grains also have significant anticancer and chemopreventive properties [3]. Accordingly, food-related biocomponents with anticancer effects have gained more and more attention in recent years [4].

Peanut (*Arachis hypogaea* L.) is an indispensable food, oil, and feed crop that is widely cultivated and consumed in the world. Peanut consumption has been reported to be associated with several health benefits due to its high nutritive value [5]. However, peanut skins are generally considered agricultural and food processing waste. Except for a small amount of peanut skin used as traditional Chinese medicine and animal feed, most of the peanut skins are removed and discarded as waste. Although peanut skins have been recognized as low-value waste by-products for decades in the peanut industry, they contain abundant polyphenolic compounds, such as flavonoids, phenolic acids, procyanidins, and anthocyanins, that exhibit in vitro and in vivo antioxidant, anti-inflammatory, anti-melanogenesis, anti-microbial, anti-cardiovascular, and antitumor properties [6,7].

Melanoma is the most dangerous type of skin cancer and the fifth most common cancer diagnosed in the United States [8]. Although melanoma accounts for only about 1% of all skin cancer patients, it causes the majority of skin cancer-related deaths, and an estimated 57,000 people died from melanoma worldwide in 2020 [8]. Surgical excision is the main and curative treatment, which can reach an overall 5-year survival rate of over 95% for localized primary melanoma [9]. However, a miserable prognosis and less than 20% of the 5-year survival rate are observed in patients with metastatic melanoma. Recently, despite several emerging therapies, such as immunotherapies and targeted therapies, being approved to improve clinical outcomes of patients with metastatic melanoma, the identification of novel therapeutic approaches or personalized therapies for those patients who do not benefit from the current treatments remains a major unmet need [10].

Colorectal cancer (CRC) is the third most common malignancy worldwide, and annually, there are 1.8 million new patients with about 900,000 deaths, accounting for roughly 6% of all cancer deaths [11]. Although approximately 80% of patients with localized stage CRC (stage I–III) can be treated with curative surgical resection [12], the 5-year survival rate drops to only 15% for metastatic (stage IV) CRC [13]. Chemotherapy is usually the first line of defense in the treatment of CRC. Over the last few years, combinational strategies of fluorouracil-based chemotherapies with epidermal growth factor receptor (EGFR) or vascular endothelial growth factor (VEGF) inhibitors have been proven effective in treating patients with metastatic CRC. Even though the 5-year mortality rates have modestly declined over the past three decades [14], there is still an urgent need to identify new prognostic biomarkers and therapeutic strategies for CRC treatment.

Autophagy is one of the important cellular catabolic processes for the elimination of misfolded proteins and damaged organelles mediated by autophagosome-lysosome fusion and degradation in lysosomes [15]. In addition to normal physiological conditions, autophagy can also be induced by nutrient starvation, energy depletion, endoplasmic reticulum stress, and hypoxia [16]. Dysregulation of autophagy has been implicated in aging, neurodegenerative diseases, infectious diseases, and malignancies [17]. Although the role of autophagy in regulating cancer cell death or survival remains controversial, several studies have shown that autophagy is a critical mechanism in cancer therapy resistance because it could provide a survival mechanism and protect cancer cells from apoptosis in response to anticancer therapies [18,19,20]. Nevertheless, whether autophagy is involved in peanut skin extract-mediated antitumor effects remains unclear.

In the development and discovery of novel and effective anticancer agents for melanoma and CRC treatment from natural products, peanut skin extracts have shown potential antitumor efficacy in previous studies [21,22,23]. To investigate and validate the anticancer effect of peanut skin extracts in melanoma and CRC, the antiproliferative, anti-migration, and apoptosis induction activities of the peanut skin fractions were investigated in this study using melanoma cell lines A375 and B16F10, and CRC cell lines SW480 and HCT15. Interestingly, our results showed, for the first time, the induction of autophagy in melanoma and CRC cells upon peanut skin extract treatment. Furthermore, we obtained evidence illustrating the potential role of autophagy in peanut skin extract-induced cytotoxicity and apoptosis in melanoma and CRC cells. Together, our results not only clearly revealed the anticancer activities of peanut skin extracts in melanoma and CRC cells but also highlighted a pro-survival mechanism of autophagy induced by peanut skin extracts.

## 2. Results

### 2.1. Differential Cytotoxic Activities of Peanut Skin Fractions against Melanoma and CRC Cells

In order to isolate the fraction with the most potent anticancer bioactivity, we processed the skin of *Arachis hypogaea* L. using serial extractions and partitions (Figure 1). Next, to identify the most potent fractions contributing to the anticancer activity of peanut skin, the effects of the AH, AHH, AHE, AHB, and AHW fractions on cell viability were examined by MTT assays at 20 and 50 μg/mL for 72 h in A375 melanoma cells and SW480 CRC cells. The cell viability results showed that the AHE fraction demonstrated more potent cytotoxicity than the others in A375 (Figure 2A) and SW480 (Figure 2B) cells. We further purified the AHE extract with ion-exchange resin and different concentrations of methanol to obtain seven fractions, AHE-1 to AHE-7 (Figure 1). The cytotoxic activities of these fractions were also examined in melanoma and CRC cells. Among these seven fractions, AHE-2 displayed the most potent inhibitory effect on cell viability in A375 and B16F10 melanoma cells (Figure 2C,D), as well as in SW480 and HCT15 CRC cells (Figure 2E,F). Moreover, we further verified the suppressive effects of AHE and AHE-2 on cell viability in a dose-dependent manner in A375, B16F10, SW480, and HCT15 cells. Both extracts exhibited high cytotoxic activities with IC_50_ values of 35.7 μg/mL, 25.8 μg/mL, 42.1 μg/mL, and 27.2 μg/mL for AHE (Figure 2G), and 26.4 μg/mL, 22.0 μg/mL, 44.2 μg/mL, and 26.0 μg/mL for AHE-2 (Figure 2H) in A375, B16F10, SW480, and HCT15 cells, respectively. The cell viability results revealed that AHE-2 showed a more potent cytotoxic effect than AHE on A375, B16F10, and HCT15 cells. Equally important, it is noteworthy that no cytotoxicity of AHE or AHE-2 was observed at the highest dose on CCD966SK, a normal human skin fibroblast cell line, suggesting that AHE and AHE-2 exhibit a relatively selective growth inhibition against cancer cells compared to normal cells (Figure 2G,H). The differential susceptibility between normal fibroblasts and cancer cell responses to AHE and AHE-2 in cell viabilities could be attributable to the differences in the genetic background, such as the oncogenic addition on *Myc*, *KRAS*, or *MET* oncogenes or tumor suppressor gene mutations in cancer cells.

### 2.2. Antiproliferative Effects of Peanut Skin Extracts on Melanoma and CRC Cells

To investigate the effect of peanut skin extracts on cancer cell growth, we further assessed the effect of AHE and AHE-2 on cell proliferation in melanoma and CRC cells by colony formation assays. Colony formation assays showed that both AHE and AHE-2 significantly reduced colony formation in a dose-dependent manner in the A375 and B16F10 cells (Figure 3A,B). Consistent with the results in melanoma cells, AHE and AHE-2 also showed significant inhibitory effects on colony-forming activity in the CRC cells, SW480, and HCT15 cells (Figure 3C,D). These results suggested that AHE and AHE-2 could effectively inhibit cancer cell proliferation in melanoma and CRC cells. In order to evaluate whether the antiproliferative effects of AHE and AHE-2 were mediated through cell cycle arrest, the cell cycle distributions were analyzed after exposure of A375 cells to AHE and AHE-2 for 48 h (Figure 3E). Flow cytometry analyses demonstrated that the percentages of the S phase after AHE and AHE-2 treatment significantly increased from 12.36 ± 1.81% (DMSO) to 20.33 ± 1.84% (60 μg/mL) and 12.30 ± 0.95% (DMSO) to 19.37 ± 1.50% (60 μg/mL), respectively, coupled with a decrease in the proportion of cells in the G1 phase (Figure 3F). These results showed that exposure to AHE and AHE-2 can induce a significant cell cycle arrest in the S phase. In addition to S-phase arrest, the percentages of the sub-G1 phase were also increased in a dose-dependent manner after AHE and AHE-2 treatment, suggesting the activation of apoptosis in AHE- and AHE-2-treated A375 cells (Figure 3F).

During cell cycle progression, the Cyclin D/CDK4/CDK6/Retinoblastoma (Rb)/E2F1 pathway and the expression of tumor suppressor p53 are recognized as key regulators of cell growth and proliferation [24]. Therefore, to further confirm the antiproliferative activities of AHE and AHE-2, Western blotting was used to determine the expressions of cell cycle-related markers after the AHE and AHE-2 treatments. As shown, the expression levels of CDK4, CDK6, phosphorylation of Rb (p-Rb), E2F1, and Cyclin A were markedly reduced, and p53 expression was significantly induced in AHE- and AHE-2-treated A375 cells (Figure 3G). Reminiscent with the results in A375 cells, AHE and AHE-2 also inhibited the expressions of the above cell cycle-related proteins, CDK4, CDK6, p-Rb, E2F1, Cyclin A, and Cyclin D1, and stimulated p53 expression in a dose-dependent manner in the SW480 cells (Figure 3H). Together, these data suggested that AHE and AHE-2 effectively inhibited cancer cell proliferation in melanoma and CRC cells.

### 2.3. Melanoma and CRC Cells Undergo Apoptosis in Response to AHE and AHE-2

Pursuant to the above results, the AHE and AHE-2 treatments significantly increased the expression of p53, which can contribute to cell cycle arrest and apoptosis. To further evaluate whether the growth-inhibitory activities of AHE and AHE-2 against melanoma and colon cancer cells were associated with the induction of apoptosis, the Annexin V-FITC/propidium iodide (PI) double-staining assay was used to detect the apoptotic cells by flow cytometry. The flow cytometry results showed that the populations of apoptotic cells, including early and late apoptosis, Annexin V-FITC^+^/PI^−^, and Annexin V-FITC^+^/PI^+^, were induced in a concentration-dependent manner after treatment with AHE and AHE-2 in A375 and B16F10 cells (Figure 4A,B). In addition, the remarkable inductions of apoptosis were also observed in SW480 and HCT15 cells treated with AHE and AHE-2 (Figure 4C,D). Supporting the evidence on the role of AHE and AHE-2 in inducing apoptotic cell death in melanoma and CRC cells, treatment of the A375 and SW480 cells with AHE and AHE-2 led to apoptosis, as manifested by the increased levels of poly(ADP-ribose) polymerase-1 (PARP-1) cleavage, cleaved caspase 9, and caspase 3 (Figure 4E,F). These results indicated that AHE and AHE-2 were able to induce caspase-mediated apoptotic cell death in melanoma and CRC cells.

### 2.4. Peanut Skin Extracts Significantly Inhibit Cell Migration in Melanoma and CRC Cells

Metastasis is the hallmark of cancer and the major cause of death in patients with melanoma and colorectal carcinoma [25,26]. Therefore, we evaluated the effects of AHE and AHE-2 extracts on the migration ability of highly metastatic B16F10 melanoma cells and SW480 CRC cells with wound-healing assays. When B16F10 cells were incubated with various concentrations of AHE and AHE-2 for 12 or 15 h, the wound-healing area varied, as shown in Figure 5A,B, respectively. The wound-healing results showed that the B16F10 cells treated with 20, 40, or 80 μg/mL AHE, as well as AHE-2, exhibited significantly reduced wound closure rates compared with the control DMSO treatment (Figure 5A,B). Similarly, the wound closure rates were also significantly inhibited by treating with AHE (Figure 5C) and AHE-2 (Figure 5D) in SW480 cells. Moreover, the transwell migration assay was performed to further confirm the effect of AHE and AHE-2 on cell migration in melanoma and CRC cells (Figure 5E). As shown, the transwell results showed that AHE and AHE-2 could significantly suppress B16F10 and SW480 cell migration in a dose-dependent manner compared with the untreated group (Figure 5F). These data indicated that AHE and AHE-2 could inhibit cell migration in melanoma and CRC cells.

### 2.5. Inhibition of Autophagy Enhances AHE- and AHE-2-Induced Cytotoxicity and Apoptosis in Melanoma and CRC Cells

Autophagy plays an evolutionarily conserved and critical role in maintaining cellular homeostasis [16]. Dysregulation of autophagy has been implicated in several pathologies, including cancers [27,28]. Although autophagy may act as a tumor suppressor at the early stage of tumorigenesis or promote cancer cell death in response to stressful conditions, several studies have also demonstrated that induction of autophagy could enhance drug resistance in anticancer therapies [29,30,31]. We next examined whether the anticancer effects of AHE and AHE-2 are associated with autophagy and investigated the role of autophagy in AHE- and AHE-2-mediated cell death in melanoma and CRC cells. The GFP-LC3 (microtubule-associated protein light chain 3) puncta formation assay, a surrogate measure of autophagy induction, indicated that AHE- and AHE-2-treated A375 cells (Figure 6A) and SW480 cells (Figure 6B) displayed an increase in GFP-LC3-labeled autophagosomes in the cytoplasm compared with the control DMSO treatment. Moreover, Western blot analysis of the conversion of LC3B-I to LC3B-II, an important marker for autophagy, further confirmed that AHE and AHE-2 effectively induced autophagy in a dose-dependent manner in melanoma and CRC cells (Figure 6C).

In order to investigate the role of autophagy in AHE- and AHE-2-induced cytotoxicity, we then examined the effect of AHE and AHE-2 alone or in combination with chloroquine, an autophagy inhibitor, on cell viability in A375 and SW480 cells. As shown in Figure 6D,E, concomitant treatment with chloroquine significantly enhanced the suppressive effects of AHE and AHE-2 at 10, 20, 40, and 60 μg/mL, and at 20 and 40 μg/mL on cell viabilities in the A375 and SW480 cells, respectively. To determine whether the augmenting effects of chloroquine on AHE- and AHE-2-mediated suppression of cell viability reflected enhanced apoptotic cell death, double-staining of Annexin V-FITC/propidium iodide (PI) followed by flow cytometry analysis was performed to examine the induction of apoptosis. In combination with chloroquine, the flow cytometry results indicated that the percentages of apoptotic cells were significantly increased in A375 and SW480 cells treated with 20 and 40 μg/mL AHE or AHE-2 (Figure 6F). Together, these results suggested that autophagy could be induced and act as the survival mechanism after treatment with AHE and AHE-2 in melanoma and CRC cells.

### 2.6. The Chemoprofiling of AHE-2

Pursuant to the above findings, we tried to analyze the chemoprofiling of AHE-2 using HPLC. The fingerprint of AHE-2 is shown in Figure 7, which reveals that AHE-2 contains three major components with retention times of 5 min, 26 min, and 50 min, and one major component with a retention time of 50 min when using the detection wavelengths of 280 nm and 254 nm, respectively. The study to identify the major compounds from AHE-2 responsible for the anticancer effects of peanut skin extracts is currently underway.

## 3. Discussion

Phytochemicals have been a fruitful source of bioactive compounds, which are intensively investigated to elucidate the potential therapeutic or preventive benefits for various diseases such as cardiovascular, diabetes, hypertension, reproductive, infection, and cancer [32,33,34]. These bioactive compounds possessed diverse biological effects, including antioxidant, antibacterial, antiviral, anti-inflammatory, and antitumor activities [35,36]. In recent years, several studies have reported the promising benefits of compounds from natural products against cancer cell proliferation, angiogenesis, and metastasis, of which natural compounds have emerged as potential therapeutic agents for cancer treatment [37]. The specific molecular mechanisms underlying the antitumor activities of natural compounds are involved in inhibiting proliferation and carcinogenesis, inducing cell cycle arrest and apoptosis, increasing antioxidant status, and regulating anticancer immune responses [38]. Although conventional chemotherapy is still one of the most widely used treatments for all kinds of cancers, drug resistance, severe side effects, and lack of selective toxicity have restricted the use of these synthetic anticancer drugs. Therefore, the most currently used new potential anticancer agents are derived from natural sources [39]. Our findings in this study have demonstrated the anticancer potential of peanut skin extracts against melanoma and CRC cells. Further extensive investigation is required to identify and mechanistically evaluate the potential compounds with anticancer properties within these extracts. Furthermore, a series of inevitably pharmacological and toxicological studies have to be conducted before its clinical application.

The cell cycle and cell proliferation are precisely controlled by cyclin proteins and their enzymatic partners, the cyclin-dependent kinases (CDKs). The cyclins and CDKs form complexes to activate the activities of CDKs that control the cell cycle progression via phosphorylation of the target proteins, such as tumor suppressor protein Rb [40]. An increasing number of studies have indicated that several natural compounds suppress cell proliferation by regulating the cell cycle and leading to cell cycle arrest, which may be attributed to the downregulation of cyclins, CDKs, and the upregulation of CDK inhibitors [38]. The formation of the active Cyclin D/CDK4/CDK6 complex plays critical roles in cell proliferation and driving cell cycle G1 to the S phase through phosphorylation and inactivation of the tumor suppressor protein Rb, which in turn facilitates the dissociation of the E2F1 transcription factor and the activation of the necessary genes for S phase-specific Cyclins and of enzymes required for DNA replication, such as Cyclin A, Cyclin E, and DNA polymerase [41]. Our data indicated that treatment of both melanoma and CRC cells with AHE and AHE-2 extracts resulted in significant suppression of several cell cycle regulator expressions, including CDK4, CDK6, p-Rb, E2F1, and Cyclin A, which in turn inhibited cell proliferation and arrested the cell cycle at the S phase. In addition to cell cycle arrest, the AHE and AHE-2 extract treatments also induced apoptosis, as demonstrated by the PARP-1 cleavage and activation of caspase 9 and caspase 3. The p53 protein, a well-known tumor suppressor, is a transcription factor and can be activated in response to various cellular stress, such as DNA damage and hypoxia, to transactivate target genes involved in cell cycle arrest or apoptosis [42]. The expressions of p53 were significantly increased in AHE- and AHE-2-treated cancer cells, suggesting that p53 signaling might be involved in AHE- and AHE-2-induced cell cycle arrest or apoptosis. Our findings were in line with a previous study, which showed that peanut skin procyanidins induced cell cycle arrest at the S phase and apoptotic cell death by activating p53 in prostate cancer cells [21].

Metastasis is the leading cause of cancer recurrence and mortality. Patients with distant metastasis were associated with a worse prognosis. The 5-year survival rates are less than 15% in both melanoma and CRC patients with metastasis [43,44]. Therefore, it remains imperative to develop effective therapeutic agents that can prevent or reduce metastasis to improve the prognosis of cancer patients. Tumor metastasis is an exceedingly complex process, and the genetic and mechanistic determinants remain incompletely understood. It has been suggested that the epithelial–mesenchymal transition (EMT) plays a critical role that allows epithelial cells to lose cell–cell adherence and acquire mesenchymal properties to develop cancer invasion, migration, and metastasis [45]. The mesenchymal markers, such as N-cadherin, Vimentin, Fibronectin, Snail, Slug, and Twist, promote tumor cells to become more malignant and increase their invasiveness and metastatic activity. In this study, we provide evidence showing that AHE and AHE-2 extracts could significantly inhibit the migration capacities of melanoma and CRC cells. The mechanism underlying the anti-migration effect of AHE and AHE-2 extracts on melanoma and CRC cells warrants further investigation of whether these two extracts affect EMT signaling or mesenchymal markers. Although there is no evidence that compounds from peanut skin show effective anti-migration activity, resveratrol, one of the stilbenes in peanut skin, has been shown to have the capability to inhibit the migration of several cancer cells. Therefore, resveratrol might be one of the potential compounds in AHE and AHE-2 to inhibit the migration activities of melanoma and CRC cells.

Autophagy is a highly regulated and conserved self-degradative and recycling cellular process that delivers misfolded or aggregated proteins and damaged organelles to lysosomes through the formation of double-membraned autophagosomes [46]. Under normal conditions, basal autophagy is constitutively active and functions as an internal quality control system to maintain cellular viability and homeostasis by recycling or eliminating useless and damaged materials. Autophagy can also be induced for adaption and survival to environmental stimuli, such as nutrient starvation, hypoxia, endoplasmic reticulum (ER) stress, oxidative stress, infection, and pharmacological treatment, through the regulations of Wnt/β-catenin, AMP-activated protein kinase (AMPK), and mammalian target of rapamycin (mTOR) pathways [16,47]. Autophagy also contributes to the illness. Dysregulation of autophagy has been reported to be associated with several diseases, including cancer, neurodegenerative disorders, infectious diseases, cardiovascular diseases, diabetes, and obesity [48]. In cancer cells, autophagy plays a critical role in cancer initiation, survival, death, metastasis, drug resistance, and cancer stem cell maintenance [47]. Although emerging studies have shown that autophagy represents a potential target for cancer therapy, there is still an ongoing debate regarding whether autophagy exerts a preventive effect against cancer cells or a cytoprotective effect in response to multiple stresses or anticancer therapies. Recent studies suggested that the exact roles of autophagy in cancers are highly dependent on the cellular context, environmental status, disease stage, and external stimuli [16]. In line with the premise, autophagy is activated and provides cytoprotective effects in the advanced stage of melanoma, but not in the early stage of melanoma, to survive in a stressful microenvironment, indicating that autophagy inhibition can be an effective therapeutic strategy for metastatic melanoma [49]. Reminiscent of the role of autophagy in melanoma, elevated autophagy activity enhanced the cancer stemness and aggressiveness in colorectal cancer, and inhibition of autophagy augmented the anticancer effects of chemotherapies against colorectal cancer [50,51]. Despite the paradoxical dual role of autophagy in tumor suppression and promotion in some cancers, several anticancer clinical trials still focus on targeting autophagy with autophagy inhibitors, either alone or in combination with chemotherapies or targeted anticancer therapies [52]. In the current study, we used chloroquine (CQ) to block autophagy and found that peanut skin extract-induced apoptosis was aggravated by autophagy inhibition. Consequently, our results indicated that autophagy induction by peanut skin extract treatment may regulate the cytoprotective effects in melanoma and CRC cells.

## 4. Materials and Methods

### 4.1. Preparation of Peanut Skin Fractions

The skin of *Arachis hypogaea* (20.5 kg) was extracted two times with 75% EtOH (60 L × 2 times). The 75% ethanol extract (AH) was dried under a reduced pressure at 45 °C. The AH extract was suspended on H_2_O (2 L) and then partitioned successively with *n*-hexane (2 L), EtOAc (2 L), and *n*-BuOH (2 L), respectively. The cytotoxic activities of the soluble layers of *n*-hexane (AHH), ethyl acetate (AHE), *n*-BuOH (AHB), and H_2_O (AHW) were examined by MTT cell viability assays. Then, the most potent EtOAc layer (AHE) was subjected to a Dianion™ ion-exchange resins column (50.0 × 9.0 cm i.d.), which was eluted in a gradient manner (30%, 50%, 75%, and 100% MeOH) to provide seven fractions (AHE-1~7).

### 4.2. Cell culture and Antibodies

The human melanoma cells A375 and mouse melanoma cells B16F10 were purchased from the Bioresource Collection and Research Center (BCRC, Hsinchu, Taiwan), and human CRC cell lines, SW480 and HCT15, were obtained from the American Type Culture Collection (Manassas, VA, USA). The human normal skin fibroblast CCD966SK cell line was a kind gift from Prof. Louis Kuoping Chao (China Medical University, Taichung, Taiwan). All the cells were maintained in the recommended growth medium with 10% fetal bovine serum (FBS) (Thermo Fisher Scientific, Carlsbad, CA, USA) at 37 °C in a humidified incubator containing 5% CO_2_. The antibodies used in this study and their sources were as follows: Cyclin A, Cyclin D1, p53, β-actin, and GAPDH (Santa Cruz Biotechnology, Santa Cruz, CA, USA); p-Rb, Rb, E2F1, CDK4, CDK6, PARP-1, Caspase 3, Caspase 9, and LC3B (Cell Signaling, Beverly, MA, USA).

### 4.3. Cell Viability Assay

The effect of peanut skin fractions on cell viability was determined using 3-(4,5-dimethylthiazol-2-yl)-2,5-diphenyltetrazolium bromide (MTT) assays. Cells were seeded in 96-well plates at a density of 5000 cells/well in the presence of 10% FBS. After an overnight incubation, the cells were treated with tested fractions at the indicated concentrations, with the final DMSO concentration at 0.1% in the presence of 5% FBS for 72 h. After treatment, MTT (Biomatik, Wilmington, DE, USA) was added to each well and incubated for an additional 1 h, and then the medium was removed and replaced with DMSO to dissolve the MTT dye for a subsequent colorimetric measurement of absorbance at 570 nm. Cell viabilities were expressed as percentages of viable cells relative to the DMSO-treated control group.

### 4.4. Colony Formation Assay

Cells were seeded in 6-well plates at a density of 2 × 10^3^–1 × 10^4^ cells per well, left to attach overnight, and then incubated with a medium containing tested fractions for 7–14 days until the colonies were visible. The colonies were fixed with 4% formaldehyde (Sigma-Aldrich, St. Louis, MO, USA) and stained with crystal violet (5 mg/mL in 2% ethanol, Sigma-Aldrich). Colonies containing more than 50 cells were counted and quantified using ImageJ software (Version 1.52a). Cell proliferation is expressed as a percentage and was determined from the number of colonies.

### 4.5. Cell Cycle Analysis

Cells were seeded in 6-well plates at a density of 1 × 10^5^–2 × 10^5^ cells per well. After an overnight incubation, the cells were exposed to the tested fractions at the indicated concentrations for 48 h. After treatment, the cells were harvested and then fixed in ice-cold 70% ethanol overnight at 4 °C. After washing with ice-cold PBS, the cells were then stained with the propidium iodide (PI) working solution (100 μg/mL PI and 100 μg/mL RNaseA) for 30 min at room temperature. The cell cycle distribution was analyzed using BD FACSCanto™ (BD, Franklin Lakes, NJ, USA) with BD FACSDiva™ 6.2 software (BD).

### 4.6. Annexin V-FITC/PI Apoptosis Analysis

Cells were seeded in 6-well plates at a density of 1 × 10^5^–2 × 10^5^ cells per well. After an overnight incubation, the cells were exposed to the tested fractions at the indicated concentrations, with the final DMSO concentration at 0.1% for 72 h. After treatment, the cells were collected and stained with 5 μL Annexin V-fluorescein isothiocyanate (FITC) and 5 μL Propidium Iodide (PI) in binding buffer for 15 min according to the manufacturer’s instructions (BD Pharmingen, San Diego, CA, USA). The stained cells were then analyzed by using BD FACSCanto™ (BD, Franklin Lakes, NJ, USA) with BD FACSDiva™ 6.2 software (BD) for the acquisition and analysis.

### 4.7. Western Blot Analysis

The cell lysates were extracted by RIPA lysis buffer with protease and phosphatase inhibitors. The protein concentrations were determined by the BCA Protein Assay kit (Thermo Fisher Scientific, Waltham, MA, USA). Protein samples were separated by SDS-PAGE and then electrophoretically transferred onto nitrocellulose membranes. The transferred membranes were blocked with 5% non-fat milk for 1 h and then incubated with primary antibodies overnight at 4 °C. After washing three times with TBST, the membranes were incubated with the corresponding secondary antibodies for 1 h at room temperature. Chemiluminescence Reagent Plus (Perkin-Elmer; Waltham, MA, USA) was used to detect signals. The expression of each protein band was quantified by densitometry analysis using ImageJ software (Version 1.52a).

### 4.8. Scratch Wound-Healing Assay

Cells were seeded in 6-well plates at a density of 4 × 10^5^–6 × 10^5^ cells per well. When the cells reached confluence, a sterile tip was used to make scratch wounds across each well. Each well was washed with PBS, and the cells were then incubated with a medium containing the tested fractions at the indicated concentrations. Images per scratch were taken by the optical microscope (magnification ×40) at 0, 12, 15, 24, or 30 h. The residual wound area was evaluated and quantified using ImageJ software (Version 1.52a).

### 4.9. Transwell Migration Assay

The migration ability of the cells was determined using transwell chambers (8 µm pore-size, Millipore, Bedford, MA, USA). The 3 × 10^4^ cells were suspended in 300 μL of medium without FBS and added to the upper chambers of each transwell insert in triplicate. An amount of 800 μL of culture medium with 10% FBS (Thermo Fisher Scientific) was added to the lower chambers, and the cells were cultured in a humidified incubator at 37 °C with 5% CO_2_. After 16~24 h, transwell inserts were fixed in methanol, stained with Giemsa, photographed, and the migrated cells were counted. Cell numbers in randomly selected fields of each independent experiment were counted using a light microscope at 100× magnification. The migrated cell numbers were expressed as percentages relative to the corresponding DMSO-treated control group.

### 4.10. Transient Transfection and GFP-LC3 Fluorescence Imaging Analysis

Cells were seeded onto round cover glasses in 6-well culture plates. GFP-LC3 plasmid (Addgene, Cambridge, MA, USA) was transiently transfected into the cells using Lipofectamine 2000 (Thermo Fisher Scientific) according to the manufacturer’s protocol. After the transfection for 24 h, the cells were incubated with DMSO or the tested fractions for another 48 h, and then the cells were fixed with 4% paraformaldehyde (in PBS) for 20 min. After washing with PBS, the nuclei were stained with 4′,6-diamidino-2-phenylindole (DAPI), and then the cover glasses were mounted with VECTASHIELD^®^ mounting medium (Vector Laboratories, Burlingame, CA, USA). GFP-LC3 autophagosome was observed, and images were recorded under a fluorescence microscope (Nikon ECLIPSE 80i, Tokyo, Japan) equipped with a DS-Qi1MC CCD camera (Nikon).

### 4.11. HPLC Analysis

HPLC profiles of the AHE-2 fraction were determined using an RP-C18 column (Cosmosil, 5C18-AR-II, 4.6 mm × 250 mm; 5 μm). HPLC was performed in the LC-20AT HPLC System (Shimadzu Co., Kyoto, Japan) equipped with an SPD-M20A photodiode array detector, of which the temperature was 25 °C. The gradient elution procedure of mobile phase A (0.05% trifluoroacetic acid in water, ≥99.0%, Merck, Darmstadt, Germany) and mobile phase B (methanol, ≥99.0%, Merck, Darmstadt, Germany) was as follows: 0–68 min, 10–100% B at a flow rate of 1.0 mL/min. Fraction AHE-2 was prepared by dissolving it in methanol (1.0 mg/mL), filtered through a 0.22 μm membrane filter, and applied for the HPLC analysis. Ultraviolet exposures at 254 and 280 nm were used.

### 4.12. Statistical Analysis

Most of the results in this study were presented as means ± standard deviations (S.D.). The statistical analyses in this study were performed by using GraphPad Prism version 8 (Graphpad, San Diego, CA, USA). A comparison between different groups was carried out using the Student’s *t*-test or one-way analysis of variance (ANOVA) followed by Tukey’s multiple comparison test. The difference is considered significant when *p* < 0.05.

## 5. Conclusions

Although peanut skin has been recognized as a low-value by-product of peanut manufacturing and is often discarded as waste or used for animal feed, accumulating evidence has demonstrated that peanut skin contains several bioactive compounds, such as flavonoids, phenolic acids, procyanidins, and anthocyanins, which are known for providing valuable biological activities, such as antioxidant, anti-infection, anti-inflammation, and health beneficial effects [6]. Therefore, peanut skin has been shown to have great potential to serve as an economical source of natural ingredients for improving health and wellness. In conclusion, we have shown the potent anticancer activities of peanut skin extracts, AHE and AHE-2, which inhibited cell viability, proliferation, migration, and induced apoptosis in melanoma and colon cancer cells. Among the polyphenols identified in peanut skin, procyanidins and resveratrol have been demonstrated to exhibit various anticancer activities, including proliferation inhibition, cell cycle arrest, apoptosis activation, anti-migration, and induction of autophagy, in different cancer cells. Therefore, procyanidins and resveratrol are the potential major compounds in AHE and AHE-2 extracts that can be used against melanoma and CRC cells. Moreover, our study uncovered a novel mechanism of peanut skin extracts that could induce cytoprotective autophagy in cancer cells, and its inhibition led to much more cell apoptosis. However, the detailed molecular mechanisms underlying the induction of autophagy by peanut skin extracts warrant further investigation. In addition, identifying the major compounds responsible for the anticancer effects of peanut skin extracts is currently underway.

## Figures and Tables

**Figure 1 ijms-25-01345-f001:**
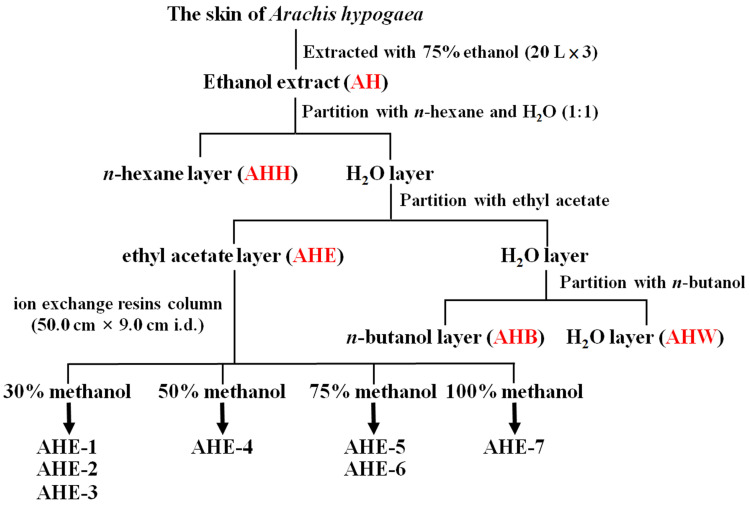
Bioassay-guided fractionation of extracts of the skin of *Arachis hypogaea*.

**Figure 2 ijms-25-01345-f002:**
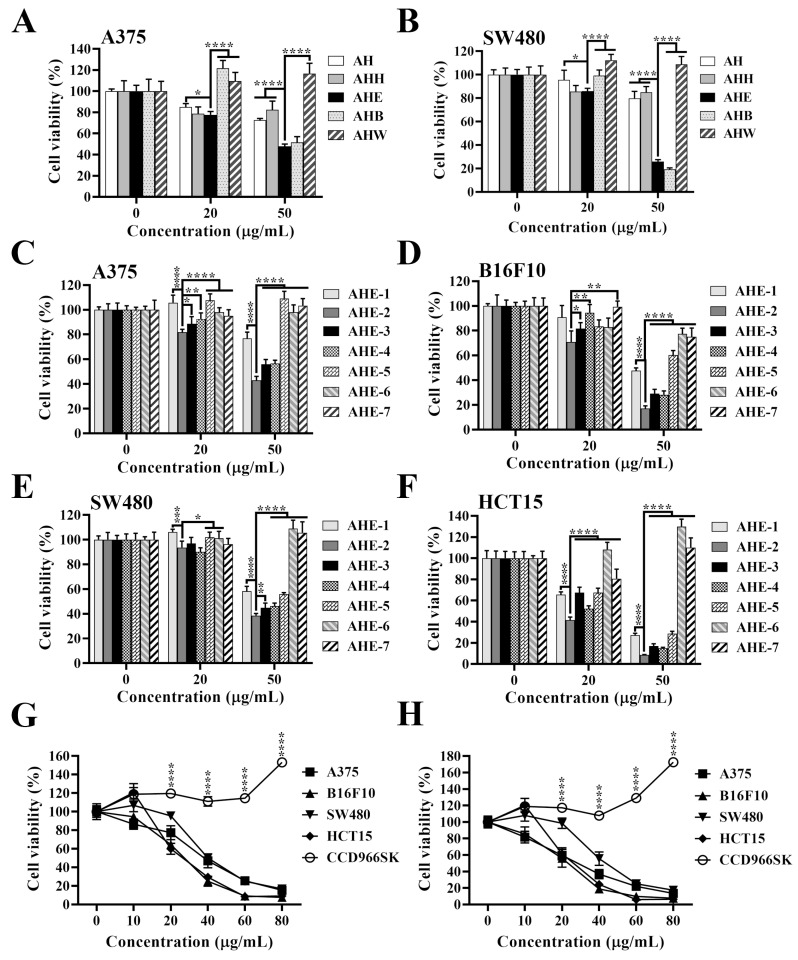
Differential cytotoxic activities of *Arachis hypogaea* skin fractions in melanoma and CRC cells. Cytotoxic effects of *Arachis hypogaea* skin extracts, including AH, AHH, AHE, AHB, and AHW, and AHE-1~AHE-7 fractions on the viability of (**A**,**C**) A375 cells, (**B**,**E**) SW480 cells, (**D**) B16F10 cells, and (**F**) HCT15 cells. Cancer cells were treated with indicated concentrations of individual extracts for 72 h, and cell viability was analyzed by MTT assay. Data are represented as the means ± S.D. (**G**) Dose-dependent suppressive effects of AHE and (**H**) AHE-2 fractions on the viability of A375, B16F10, SW480, HCT15, and CCD966SK cells after 72 h treatment. Cell viability was analyzed by MTT assay. Data are represented as the means ± S.D. from three independent experiments. * *p* < 0.05, ** *p* < 0.01, *** *p* < 0.001, **** *p* < 0.0001.

**Figure 3 ijms-25-01345-f003:**
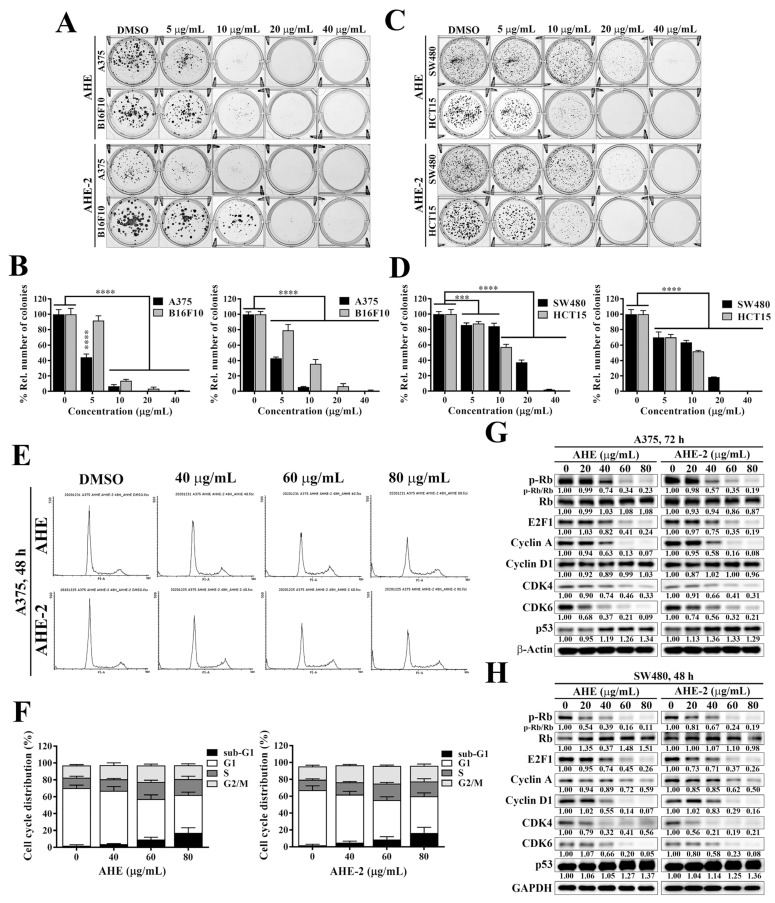
Effects of AHE and AHE-2 fractions on cell proliferation in melanoma and CRC cells. (**A**) Representative images showing the effects of AHE and AHE-2 fractions on colony formation in A375 and B16F10 cells. (**B**) Quantitative results of the effects of AHE (left) and AHE-2 (right) fractions on colony formation in A375 and B16F10 cells and expressed as the percentage compared with the DMSO control. Data are represented as the means ± S.D. Significant difference versus control: **** *p* < 0.0001. (**C**) Representative images showing the effects of AHE and AHE-2 fractions on colony formation in SW480 and HCT15 cells. (**D**) Quantitative results of the effect of AHE (left) and AHE-2 (right) fractions on colony formation in SW480 and HCT15 cells and expressed as the percentage compared with the DMSO control. Data are represented as the means ± S.D. Significant difference versus control: *** *p* < 0.001, **** *p* < 0.0001. (**E**) Flow cytometry analyses of the cell cycle distributions after treatment of AHE and AHE-2 for 48 h in A375 cells. (**F**) Quantitative results of the effect of AHE (left) and AHE-2 (right) fractions on cell cycle distributions in A375 cells. Data are represented as the means ± S.D. from three independent experiments. (**G**) Western blot analyses of the dose-dependent effect of AHE and AHE-2 fractions on the expression levels of cell cycle-related markers, including p-Rb, Rb, E2F1, Cyclin A, Cyclin D1, CDK4, CDK6, and p53, in A375 and (**H**) SW480 cells. The fold changes of protein expressions were determined by the relative intensity of protein bands of treated samples to that of the respective DMSO-treated control after normalization to the respective internal reference β-actin or GAPDH protein.

**Figure 4 ijms-25-01345-f004:**
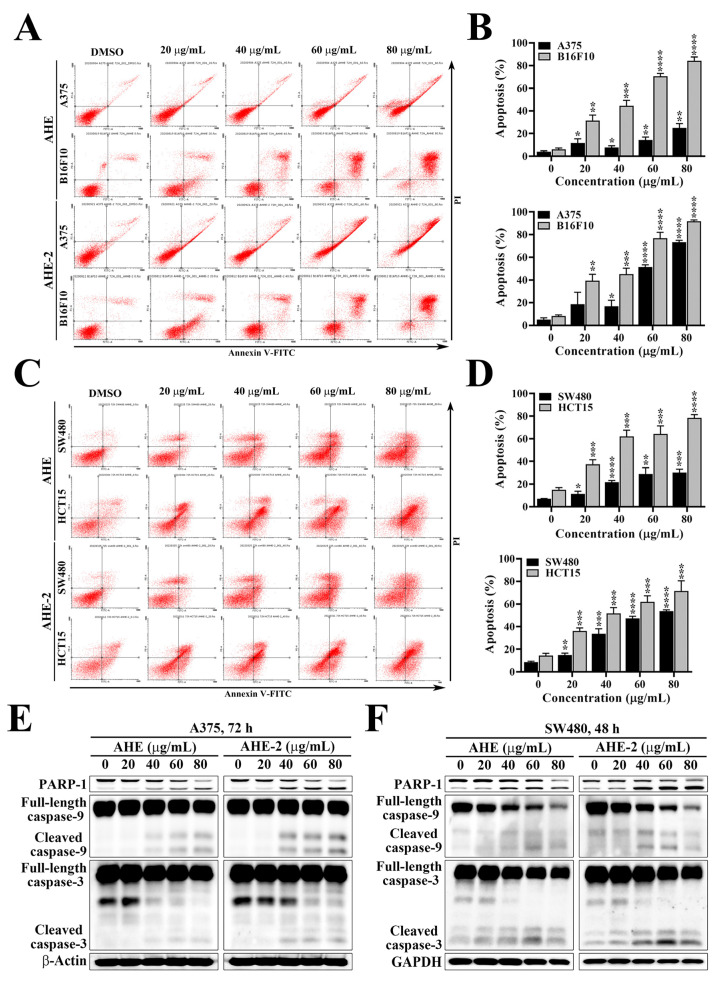
AHE and AHE-2 fractions induced apoptosis in melanoma and CRC cells. (**A**) Representative results showing the effects of AHE and AHE-2 fractions on apoptosis in A375 and B16F10 cells. Annexin V-FITC/propidium iodide double-staining of A375 and B16F10 cells after treatment with AHE and AHE-2 fractions at the indicated concentrations for 72 h. (**B**) Quantitative results of the effects of AHE (upper) and AHE-2 (lower) fractions on apoptosis in A375 and B16F10 cells and expressed as the percentage compared with the DMSO control. Data are represented as the means ± S.D. from three independent experiments. (**C**) Representative results showing the effects of AHE and AHE-2 fractions on apoptosis in SW480 and HCT15 cells. Annexin V-FITC/propidium iodide double-staining of SW480 and HCT15 cells after treatment with AHE and AHE-2 fractions at the indicated concentrations for 72 h. (**D**) Quantitative results of the effects of AHE (upper) and AHE-2 (lower) fractions on apoptosis in SW480 and HCT15 cells and expressed as the percentage compared with the DMSO control. Data are represented as the means ± S.D. from three independent experiments. * *p* < 0.05, ** *p* < 0.01, *** *p* < 0.001, **** *p* < 0.0001. (**E**) Dose-dependent effects of AHE and AHE-2 on various apoptosis biomarkers, including PARP-1 cleavage and activation of caspase 3 and caspase 9, in A375 cells and (**F**) SW480 cells.

**Figure 5 ijms-25-01345-f005:**
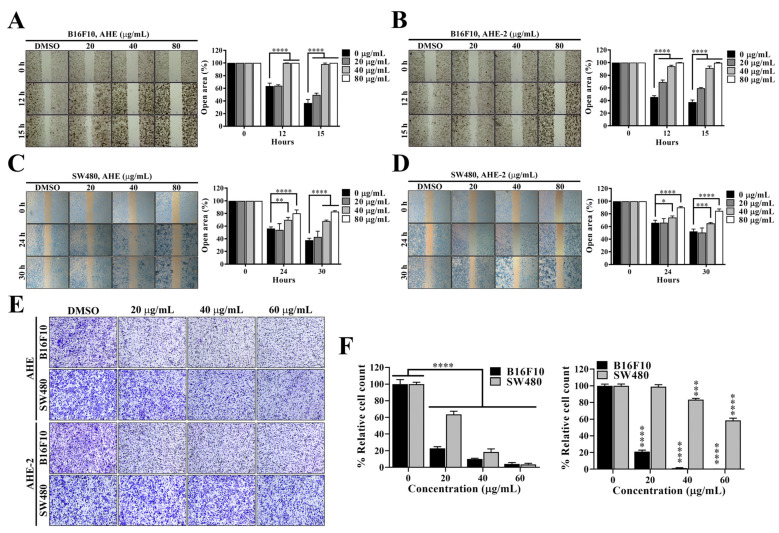
AHE and AHE-2 fractions inhibited melanoma and CRC cell migration. (**A**) Representative images (left) and quantitative results (right) showing the dose-dependent effects of AHE and (**B**) AHE-2 fractions on B16F10 cell migration by wound-healing assays (magnification ×40). Relative open area was expressed as the percentage compared with the DMSO control. Data are represented as the means ± S.D. from three independent experiments. Significant difference versus control: **** *p* < 0.0001. (**C**) Representative images (left) and quantitative results (right) showing the dose-dependent effects of AHE and (**D**) AHE-2 fractions on SW480 cell migration by wound-healing assays (magnification ×40). Relative open area was expressed as the percentage compared with the DMSO control. Data are represented as the means ± S.D. from three independent experiments. Significant difference versus control: * *p* < 0.05, ** *p* < 0.01, *** *p* < 0.001, **** *p* < 0.0001. (**E**) Representative images showing the dose-dependent effects of AHE and AHE-2 fractions on B16F10 and SW480 cell migration by transwell migration assays. Scale bar, 100 μm. (**F**) Quantitative results of transwell migration assays showing the dose-dependent effects of AHE (left) and AHE-2 (right) fractions on B16F10 and SW480 cell migration. Data are represented as the means ± S.D., *** *p* < 0.001, **** *p* < 0.0001.

**Figure 6 ijms-25-01345-f006:**
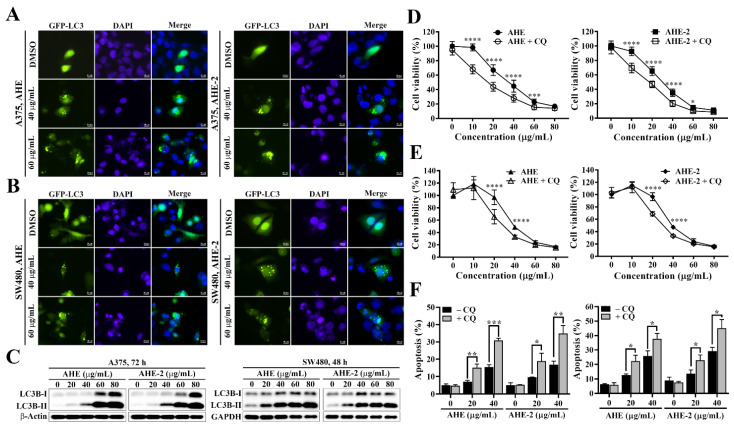
Autophagy inhibitor enhances the inhibitory effects of AHE and AHE-2 and promote the induction of apoptosis in melanoma and CRC cells. Representative fluorescence images showing the effect of AHE (left) and AHE-2 (right) on autophagy in GFP-LC3 expressing (**A**) A375 cells and (**B**) SW480 cells. Fluorescence imaging analysis of GFP-LC3 fluorescence puncta in GFP-LC3 expressing cells after treatment with AHE and AHE-2 fractions at the indicated concentrations for 48 h. Scale bar, 10 μm. (**C**) Western blot analyses of the dose-dependent effect of AHE and AHE-2 fractions on the expression of LC3B conversion in A375 (left) and SW480 (right) cells. (**D**) Cytotoxic effects of AHE (left) and AHE-2 (right) on the viability of A375 cells and (**E**) SW480 cells in the absence or in the presence of chloroquine (CQ). Cancer cells were treated with indicated concentrations of individual fractions alone or in combination with chloroquine for 72 h, and cell viability was analyzed by MTT assay. Data are represented as the means ± S.D. from three independent experiments. Significant difference versus control: *** *p* < 0.001, **** *p* < 0.0001. (**F**) Quantitative results of the effects of AHE and AHE-2 (lower) fractions on apoptosis in A375 cells (left) and SW480 cells (right) in the absence or in the presence of chloroquine (CQ) and expressed as the percentage compared with the DMSO control. Data are represented as the means ± S.D. from three independent experiments. Significant difference versus control: * *p* < 0.05, ** *p* < 0.01, *** *p* < 0.001.

**Figure 7 ijms-25-01345-f007:**
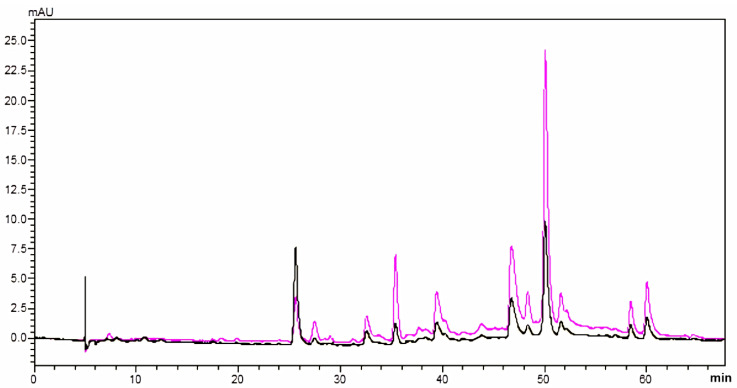
HPLC chromatogram analysis of AHE-2 fraction (pink: 254 nm; black: 280 nm).

## Data Availability

All the data of this research are available upon reasonable request.

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
