# Peer review of "Inhibition of Autophagy Aggravates Arachis hypogaea L. Skin Extracts-Induced Apoptosis in Cancer Cells"

_ijms, 2024, doi:10.3390/ijms25021345_

Round 1

Reviewer 1 Report

Comments and Suggestions for Authors

This manuscript investigated the impact of Arachis hypogaea L. (peanut or groundnut) skin extracts on cancer cells. The authors observed that the ethyl acetate fraction (AHE) of peanut skin ethanolic crude extract and one of the 30% methanolic fraction (AHE-2) from ethyl acetate extraction inhibited the proliferation of melanoma and colorectal cancer cells. Furthermore, these extracts arrested the cell cycle and induced apoptosis in these cells. The authors demonstrated that the cytotoxicity induced by AHE and AHE-2 was potentiated by autophagy inhibition. This manuscript provides novel insights into the influence of peanut skin extracts on cancer cell proliferation. The following points should be addressed.

1.       The authors should discuss the mechanisms underlying the action of peanut skin extracts in cancer cells.

2.       The authors should discuss the anticipated compounds present in the peanut skin extracts.

3.       Line 408: "in some3" should be corrected to "in some."

Reviewer 2 Report

Comments and Suggestions for Authors

This manuscript, “Inhibition of Autophagy Aggravated Arachis hypogaea L. Skin Extracts-induced Apoptosis in Cancer Cells,” by Tsai et al., is very well written and analyses the effects of peanut skin extracts for anti-tumor activities. The authors extracted polyphenols from peanut skin and systematically showed the antiproliferative activity on melanoma and colon cancer cell (CRC) lines. The authors have analyzed the effect of peanut skin polyphenols AHE and AHE2 in multiple melanoma and colon cancer cell lines along with a normal skin cell line. The effect of AHE and AHE2 on the cell cycle has been shown to affect the S-phase and several key cell cycle arrest markers. Later, the authors showed that AHE and AHE-2 induce apoptosis in melanoma and CRC cells. The authors used migration assay to show that both polyphenols inhibited cell migration. Last but not least, the authors have shown that both polyphenols induce autophagy using a GFP-LC3 puncta formation assay. Though the manuscript is written well, the authors should consider the mentioned suggestions to improve its impact further.

1.     It would be nice to see if other polyphenols (natural or synthetic) have similar effects to those from peanut skin. If yes, then why, or if not, then why not?

2.     The authors mentioned that the differences in AHE and AHE-2 activity on cell proliferation could be due to genetic background without any explanation; it would be nice to elaborate on this point.

3.     Please consider mentioning different cell cycle phases in the figure 3E. Also, the figure is of low resolution. Please consider using a high-resolution figure.

4.     Fig 3G & 3H, please consider providing supplementary WB quantification.

5.     For Fig 5C & 5D, please consider changing the figure to higher resolution ones.

6.     Why is only the AHE2 chemoprofile shown? Why not AHE? Please consider labeling Figure 7 and mentioning all the relevant peaks.

7.     Please consider rewriting the discussion. The opening of the discussion seems like it is being copied from the introduction. Additionally, focus on discussing the results more, along with justification for the different aspects being tested.  Please consider projecting what should be done next to further enhance and develop natural polyphenols as chemotherapeutic drugs.

Thank you,

Best

US

Reviewer 3 Report

Comments and Suggestions for Authors

The manuscript "Inhibition of Autophagy Aggravated Arachis hypogaea L. Skin Extracts-induced Apoptosis in Cancer Cells" is a novel,  well-planned research design and conducted manuscript. Some minor suggestions are as follows:

·       In Fig. 1 &4, the author should perform statistical analysis.

·       Check the entire manuscript for minor grammatical mistakes and clear and concise language. 

Comments on the Quality of English Language

None

Round 2

Reviewer 1 Report

Comments and Suggestions for Authors

The authors have satisfactorily addressed the points that I previously noted.